# Ru(II) Oxygen Sensors for Co(III) Complexes and Amphotericin B Antifungal Activity Detection by Phosphorescence Optical Respirometry

**DOI:** 10.3390/ijms24108744

**Published:** 2023-05-14

**Authors:** Katarzyna Turecka, Agnieszka Chylewska, Aleksandra M. Dąbrowska, Rafał Hałasa, Czesława Orlewska, Krzysztof Waleron

**Affiliations:** 1Department of Pharmaceutical Microbiology, Faculty of Pharmacy, Medical University of Gdańsk, al. Hallera 107, 80-416 Gdańsk, Poland; rafal.halasa@gumed.edu.pl (R.H.); krzysztof.waleron@gumed.edu.pl (K.W.); 2Department of Bioinorganic Chemistry, Faculty of Chemistry, University of Gdańsk, Wita Stwosza 63, 80-308 Gdańsk, Poland; agnieszka.chylewska@ug.edu.pl (A.C.); aleksandra.dabrowska@ug.edu.pl (A.M.D.); 3Department of Organic Chemistry, Faculty of Pharmacy, Medical University of Gdańsk, al. Hallera 107, 80-416 Gdańsk, Poland; corl@gumed.edu.pl

**Keywords:** oxygen sensors, Co(III) complexes, phosphorescence optical respirometry, antifungal activity, *Candida albicans*

## Abstract

The measurement of oxygen consumption is an important element in the understanding of an organism’s metabolic state. Oxygen is also a phosphorescence quencher, which allows the evaluation of phosphorescence emitted by oxygen sensors. Two Ru(II)-based oxygen-sensitive sensors were used to study the effect of chemical compounds [(**1**) = [CoCl_2_(dap)_2_]Cl, and (**2**) = [CoCl_2_(en)_2_]Cl (AmB = amphotericin B) against reference and clinical strains of *Candida albicans*. The *tris*-[(4,7-diphenyl-1,10-phenanthroline)ruthenium(II)] chloride ([Ru(DPP)_3_]Cl_2_) (Box) adsorbed onto the Davisil^TM^ silica gel was embedded in the silicone rubber Lactite NuvaSil^®^ 5091 and the coating on the bottom of 96-well plates. The water-soluble oxygen sensor (BsOx = *tris*-[(4,7-diphenyl-1,10-phenanthrolinedisulphonic acid disodium)ruthenium(II)] chloride ‘x’ hydrate = {Ru[DPP(SO_3_Na)_2_]_3_}Cl_2_ = water molecules were omitted in the BsOx formula) was synthesized and characterized by RP-UHPLC, LCMS, MALDI, elemental analysis, ATR, UV-Vis, ^1^H NMR, and TG/IR techniques. The microbiological studies were performed in the environment of RPMI broth and blood serum. Both Ru(II)-based sensors turned out to be useful in the study of the activity of Co(III) complexes and the commercial antifungal drug amphotericin B. In addition, a new activity of the oxygen sensor, the soluble Ru(II) complex BsOx, was demonstrated, which is a mixture with amphotericin B that caused a significant increase in its antifungal activity. Thus, it is also possible to demonstrate the synergistic effect of compounds active against the microorganisms under study.

## 1. Introduction

Analysis of the cell population in respect of their viability is an essential test in the fields of cell biology, biotechnology, and the processes of discovering new drugs, including antimicrobial compounds. A significant number of tests have been developed for evaluating viability in a more convenient, automated, and economical way. They were achieved using relatively simple probes, microplates, and appropriate detectors, while the analysis itself was based on measuring the phosphorescence emission of the sample. Oxygen is a key metabolite of all aerobic biological systems, and oxygen consumption provides useful information about the metabolic activity of cells. This element is also an effective phosphorescence quencher, which facilitates the conductance of analysis and testing of cell viability using so-called optical respirometry based on phosphorescence [1,2,3]. This method allows the measurement of phosphorescence emitted by an oxygen-sensing probe, the emission of which depends on the amount of oxygen in the test sample. The process leading to the reduction in phosphorescence intensity is called “phosphorescence quenching”. Molecular oxygen colliding with an excited fluorophore molecule (a part of the molecule responsible for its phosphorescence) quenches the phosphorescence and causes the fluorophore to return to its ground state so it is not able to emit radiation. As a result, the intensity of luminescence and its viability in the presence of oxygen decrease [1,4,5]. The method of phosphorescence quenching for the measurements of oxygen in live tissue and biological samples was introduced by Rumsey et al. [6,7]. Oxygen-sensing probes have been used for the analysis of mammalian cells [8], isolated mitochondria [9], small organisms [10] or aerobic bacteria and yeasts [11,12,13]. This method tested organisms growing, and consuming oxygen, resulting in an effect on the phosphorescence intensity of the sample. The biosensor signal is quenched by oxygen in a reversible, non-chemical way, and oxygen depletion/the reduction in the concentration of dissolved oxygen (which is associated with microbial growth and aerobic respiration) causes a significant increase in biosensor phosphorescence emission. Monitoring of the changes in phosphorescence intensity in cultures allows us, therefore, to observe the metabolic activity of organisms, and the influence of physical factors and chemical compounds on organisms’ activity in real-time [4,11,12]. This method is rapid, sensitive, and useful for performing studies based on multiple samples [12]. Analyzes of this type have been applied in the toxicological evaluation of chemical and biological samples in various scientific areas. They are helpful when creating new drugs, monitoring the environment, or examining animal physiology [10].

One of the most valued and widely used oxygen biosensors is fluorescent complexes of Ru(II) [14]. The [Ru(DPP)_3_]Cl_2_ in PDMS [poly(dimethylsiloxane)] is characterized by an excitation optimum at 460 nm, an emission optimum at 610 nm, and a lifetime of about 4 µs [15]. Moreover, it is highly stable and can even be steam-sterilized [16]. In turn, water-soluble {Ru[DPP(SO_3_Na)_2_]_3_}Cl_2_ is characterized by a lifetime of 3.7 μs, excitation optimum at 480 nm, emission optimum at 615 nm, and stability in aqueous solutions, and was used for research on eukaryotic cell cultures and yeasts [17].

The ruthenium-based phosphorescent oxygen-sensitive biosensor coating on the bottom of 96- or 384-well plates was introduced by Wodnicka and co-workers [18] and developed by Becton Dickinson Biosciences as “The Oxygen BioSensor”. The usefulness of this system has been widely described by us in our previous paper. We showed that these assays allow real-time cell growth monitoring on a microscale, allow the time to obtain results to be significantly reduced (from 24–48 h to 5–7 h for bacteria), and control experiments showed the culture volume can be also reduced (from 160 μL to 50 μL). Moreover, this system proved to be useful for analyzing the kinetics of the culture growth of bacteria and the cell growth parameters, and also for studying the antimicrobial activity of chemical compounds and determining the MIC values [12]. In turn, there are no literature data available on the use of soluble *tris*-[(4,7-diphenyl-1,10-phenanthrolinedisulphonic acid disodium)ruthenium(II)] chloride (BsOx) in studies using phosphorescence optical respirometry. The above-mentioned sensor, but embedded within 45 nm-hydrodynamic-diameter nanoparticles of polyacrylamide was used in the study of *Schizosaccharomyces pombe* and *Saccharomyces cerevisiae* yeasts cells and human cell line (mammary adenosarcoma MCF-7) for mapping of intracellular O_2_ distributions using imaging techniques [19,20,21,22].

In this study, we have examined the effect of chemical compounds with the formula [CoCl_2_(dap)_2_]Cl = (**1**), [CoCl_2_(en)_2_]Cl = (**2**), amphotericin B = AmB) against *Candida* species in RPMI medium and serum using and comparing two oxygen-sensitive sensors: *tris*-(4,7-diphenyl-1,10-phenanthroline)ruthenium(II) chloride {[Ru(DPP)_3_]Cl_2_ = Box} adsorbed on the Davisil^TM^ silica gel was embedded in the silicone rubber Lactite NuvaSil^®^ 5091, the coating on the bottom of 96-well plates [12,18], and the water-soluble oxygen sensitizer *tris*-[(4,7-diphenyl-1,10-phenanthroline-disulphonic acid disodium)ruthenium(II)] chloride (BsOx). Although Ru(II)-based sensor studies have been published previously [12,18,22], we demonstrated in our work for the first time the applicability sensors to test the antifungals (amphotericin B and Co(III) complexes) against *Candia* spp. cells. Additionally, studies using a soluble sensor, {Ru[DPP(SO_3_Na)_2_]_3_}Cl_2_ = BsOx, to test the antifungal activity of chemical compounds against *Candida albicans* strains were performed for the first time by the phosphorescence optical respirometry method. We have also not found literature data indicating the use of the aforementioned sensor in an antibacterial activity study using the phosphorescence optical respirometry method.

## 2. Results and Discussion

### 2.1. Structural Characterization of the Coordination Compounds Studied

The purity of the *tris*-[4,7-diphenyl-1,10-phenanthrolinedisulphonic acid disodium)ruthenium(II)] chloride ‘x’ hydrate (BsOx) synthesized (Figure 1) was confirmed by RP-UHPLC to be >99% (Appendix A).

The BsOx compound presents MLCT absorption bands in the visible region at about 440 and 465 nm (Appendix A) and also π → π* transitions centered on the ligand in the UV range. The absorption band at 277 nm is associated with the presence of a 4,7-diphenyl-1,10-phenanthroline disulfonic acid disodium ligand [21]. The results confirmed those obtained by Castellano et al. [17] and are also in agreement with Coogan’s study [22]. In turn, TG/IR results for the BsOx sample were presented in Appendix A. Based on those results, the hydration state of the BsOx solid form was established. The presence of 12 molecules of water outside of the coordination sphere was proved by the first weight loss (found: 9.77%; calc.: 9.91%). Interestingly, the more strongly bound water located at an intra-cation complex was confirmed by the further weight losses in the BsOx thermal decomposition (Appendix A), based also on the characteristic water oscillatory vibration bands in the IR spectra (Appendix A). The results of the MALDI-TOF analysis turned out to be consistent with those obtained by the LCMS method; the complete information on the relative intensities with specific mass-to-charge ratios and their descriptions have been compiled in the Appendix A. Additionally, the redox potentials of *tris*-[(4,7-diphenyl-1,10-phenanthrolinedisulphonic acid disodium)-ruthenium(II)] chloride measured in DMSO by using the CV method were also in good agreement with those reported in the literature for Ru(II) water-soluble oxygen sensor [21]: E_ox_ [V/SCE] = −1.81; −0.93; −0.82; −0.49; +0.57; +0.71 and +1.20; E_red._[V/SCE] = −0.85; −1.43; −1.88 (Appendix A). The potentials observed on the voltammogram prove the presence of ruthenium on the second oxidation state being an ionic metallic center of the complex. The other results of analyses for BsOx were collected in Appendix A. The structural characterizations of Box, Co(III) coordination compounds (**1**) and (**2**) were included in the Appendix A, respectively.

### 2.2. Monitoring Microbial Growth Using Phosphorescence Optical Respirometry

In this study, the reference and clinical two strains of *Candida albicans* were used. Their viability and the effect of the compounds with proven antimicrobial activity (antifungal and cobalt compounds) were evaluated using the phosphorescence optical respirometry method. The study aimed to compare both phosphorescent biosensors Box and BsOX. The most important advantage of these complexes is their long lifetime (hundreds of nanoseconds and microseconds), ability to excite in the visible range, high quantum yields, and large Stokes shift values. The presence of metal ions in the complexes (the heavy atom effect) by increasing the spin-orbit coupling facilitates inter-system transitions and the occupation of triplet levels. This allows the registration of the phosphorescence of biosensors at room temperatures. As the lifetime of excited states (triplet states) increases, the effect of the quencher also increases.

The oxygen-sensitive sensors were used to study the metabolic activity of yeasts. The determination of the optimal concentration of BsOx was carried out first (Appendix A). To check the concentration range of the BsOx sensor in which tests using optical respirometry should be conducted, the activity/toxicity studies (minimum inhibitory concentration, MIC) of this sensor against *Candida albicans* strains were performed by the microbroth dilution method. The MIC_s_ for both strains of *Candida albicans* were 1000 µg/mL, i.e., this concentration was the lowest growth-inhibiting concentration of the tested strains. Therefore, the concentration of BsOx tested was within the range of 250–7.8 µg/mL and was performed for both strains of *Candida albicans* with an optical density of 10^7^ CFU/mL (selected optical density according to the procedure described by Hałasa et al.) [12]. The best results, i.e., the shape of the fungal growth curves most similar to the sigmoidal shape characteristic for the growth of microorganisms in the stationary culture, were obtained for higher concentrations of the analyzed sensor BsOx (250, 125, and 62.5 µg/mL). The curves characterizing lower concentrations (31.25, 15.6, or 7.8 µg/mL) differed significantly in shape from typical sigmoidal curves and the interpretation of the results based on the above-mentioned graphs would not be accurate (Appendix A). Therefore, a concentration of 62.5 µg/mL was selected for further testing.

The next stage of our work was to compare the Box and BsOx sensors at studying the metabolic activity of yeasts, so fungal growth monitoring experiments were carried out (final density ranging from 10^8^ to 10^4^ CFU/mL, time 19 h). At the time studied, samples containing Box sensors showed sigmoidal shape curves of changes in the intensity of phosphorescence against time (Figure 2a). In these cases, the samples reached a plateau, which means that they reached the stationary phase (irrespective of the initial density of the yeast). However, depending on the initial fungal density, time to the maximum level of phosphorescence for the stationary phase was varied. The 1 × 10^8^ CFU/mL culture reached its maximum after 6 h, while the most diluted culture took about 18 h to reach the stationary phase. Thus, the lower the initial density of the yeast culture, the longer the time to reach the maximum level. These results are consistent with data obtained by Hałasa et al. [12] and Wodnicka et al. [18]. In the case of the BsOx sensor, the curves did not show typical sigmoidal shape curves of the changes in the intensity of phosphorescence against time (more flattened), but it was still possible to determine the time when the culture reached a plateau (except for the culture with the lowest optical density, 10^4^ CFU/mL) (Figure 2b). We suggest that the possible cause of the differences in the shape of the curves describing the growth/metabolism of fungi was the impact of the sensor on the analyzed cells associated with binding to proteins and membranes. Indeed, Coogan and co-workers have shown the uptake of the analyzed compound into the cell [22]. However, there are no literature data available describing the phosphorescence optical respirometry for this sensor. In the case of the Box embedded in silicone, there was no interaction with the tested sample, and the sensor in this form turned out to be the most durable.

### 2.3. Study of the Effect of Co(III) Coordination Compounds and Amphotericin B on Yeast by Phosphorescence Optical Respirometry

#### 2.3.1. Measurements Using Box, BsOx in RPMI Medium

One of the most important applications of MIC measurements is determining the lowest concentration for antibiotics or antifungal drugs to inhibit the growth of microorganisms. Based on the obtained MIC values, it is assessed whether a given strain is susceptible to an antimicrobial and on this basis, decisions are made about the method of treatment of the patient. Additionally, the possibility of using new chemotherapeutic agents in medicine must be preceded by research on the effects of these compounds on microorganisms.

In our work, the antifungal properties of Co(III) complexes with diamine chelate ligands [compounds (**1**) and (**2**)] against the reference and clinical strains of the *Candida albicans* were tested, and subsequentially the usefulness of Ru(II)-based sensors in testing antifungal activity was compared by the phosphorescence optical respirometry method. For comparative purposes, studies have also been conducted with amphotericin B (Figure 3). AmB belongs to the polyene class of antifungals and acts on fungal cells at two different levels: through pore formation at the cell membrane after binding to ergosterol, and its sequestration and induction of oxidative damage [23].

Depending on which of the oxygen-sensitive sensors was used in the study, different MIC values were obtained. In the case of Box, *C. albicans* ATCC 10231 and 12823 strains can grow in the presence of compound (**1**) at concentrations of 15.6 and 31.25 µg/mL, while concentrations of 62.5, 125, 250, and 500 µg/mL cause growth inhibition (Figure 3a,b). However, compound (**2**) inhibits the growth of the reference strain of *Candida albicans* at the concentrations of 62.5, 125, 250, and 500 µg/mL but the clinical strain of *C. albicans* at 125, 250, and 500 µg/mL. Therefore, the minimum inhibitory concentration (MIC) for compounds (**1**) and (**2**) against *C. albicans* ATCC 10231 is 62.5 µg/mL, and against *C. albicans* 12823 is 62.5 and 125 µg/mL for compounds (**1**) and (**2**), respectively. The control sample, in the presence of AmB, shows the growth of *Candida albicans* strains at concentrations of 0.06, 0.125, and 0.25 µg/mL and the inhibition of growth at a concentration of 0.5, 1, 2, 4, and 8 µg/mL (MIC is 0.5 µg/mL, which is consistent with serial dilution results) (Figure 3a,b and Table 1). In the case of the samples containing a BsOx sensor, the inhibition of growth of *Candida albicans* strains for compounds (**1**) and (**2**) was observed at concentrations of 250 and 500 µg/mL, so the MIC value is 250 µg/mL. This values were four and two times higher than the one obtained with the serial dilution method and with the [Ru(DPP)_3_]Cl_2_ sensor (Box) for compounds (**1**) and (**2**), respectively. The use of a sensor with a concentration of 125 µg/mL increased the MIC value for Co(III) compounds to 500 µg/mL. It seems that the presence of the Ru(II) complex at higher concentrations in a mixture with Co(III) coordination compounds reduces their activity. However, the presence of the BsOx at a concentration of 31.25 µg/mL does not affect the activity of Co(III); the MIC value remains unchanged at 62.5 µg/mL (Appendix A). The inhibition of *C. albicans* cell growth in the presence of amphotericin B was observed at concentrations of 0.125, 0.25, 0.5, 1, 2, and 4 µg/mL, so the MIC value is 0.125 µg/mL, and this value is four times lower than the MIC obtained with the Box sensor or with the serial dilution method (0.5 µg/mL, Table 1, Figure 3a,b).

The presence of the BsOx (62.5 µg/mL) in the tested mixture enhances the antifungal effect of amphotericin B, indicating that the compounds may act synergistically. The same MIC value was achieved in the presence of a sensor at a concentration of 125 µg/mL and at 31.25 µg/mL; the MIC for amphotericin B dropped below 0.06 µg/mL (Appendix A). We did not expect such an effect, a drastic decrease in the MIC value for AmB in the presence of the aforementioned sensor, or a decrease in activity in the case of the Co(III) complexes. We demonstrated synergistic (Ru(II) complex-AmB) and antagonistic (Ru(II) complex-Co(III) complexes) interactions of the reaction components. Therefore, we hypothesize that it is possible to demonstrate in this way an indicative mechanism of action of the tested compounds. Coogan et al. [22] showed that the Ru(II) coordination complex embedded within hydrodynamic diameter nanoparticles of polyacrylamide enters the analyzed cells. In our previous work [24], we showed that the newly synthesized FITC-labelled [Co(dap)_2_]Cl_2_ ([Co(dap)_2_FLU]Cl_2_) penetrated the bacterial cell membrane and was evenly distributed inside bacterial cells. Using confocal microscopy and the mentioned properties of the tested compounds, it is possible to determine their binding site in *Candida* spp. cells, to check whether, for example, they compete with each other for the binding site in the cell. Testing this hypothesis is the next goal of our research. On the other hand, Zu et al. [25] showed that the compounds [(Ph_2_phen)_2_Ru(dpp)]^2+^ and [(Ph_2_phen)_2_Os(dpp)]^2+^ (Ph_2_phen = 4,7-diphenyl-1,10-phenanthroline; dpp = 2,3-bis(2-pyridyl)pyrazine), with structures very similar to the compound analyzed by us, exhibited oxygen-mediated DNA and BSA photocleavage and significant photocytotoxicity under blue light irradiation, along with negligible activity in the dark [26]. Accordingly, it is also possible that the BsOx in the appropriate concentration, after irradiation with light with a wavelength (excitation) of 480 nm, acts as compounds used in photodynamic therapies. This effect, in relation to BsOx, will be our next research goal.

Experimental results with the Box sensor for compounds (**1**), (**2**) and both tested strains of *C. albicans* showed the typical sigmoidal shape of the curves for probe signal which differed in the position of their inflection point, with a significant shift in changes in the phosphorescence curves in time compared to a sample containing only the yeast culture (the pure culture of *C. albicans* strains reached maximum phosphorescence after 6 h, while samples containing test compounds reached maximum phosphorescence, for example, at 15.6 µg/mL after 13 h for compound (**1**) or 11 h for compound (**2**) at the same concentration, Figure 4a). This means that lower concentrations of the complexes (15.6, 31.25 µg/mL) caused a decrease in the metabolism of yeasts. For these concentrations, waveforms with lower intensity amplitudes could also be observed, compared to the phosphorescence curve characteristics of the pure bacterial culture. This effect increased with higher concentrations of tested compounds leading to complete inhibition of fungal growth. In the samples containing AmB, a shift in changes in the phosphorescence curves with time was observed for the lowest tested drug concentrations (0.06 and 0.125 µg/mL) compared to the control sample. Only at a concentration of 0.25 µg/mL was the shift from the yeast-only sample significant. Using polymeric materials as materials embedding the sensor (in our silicone work) can lead to the elimination of self-quenching effects and the achievement of high phosphorescent signals, as well as a reduction in possible cross-sensitivity effects.

For samples containing the BsOx sensor, it was also possible to determine the MIC value; however, it was not possible to accurately determine the moment in which a significant increase in the density of breeding occurred (Figure 3a,b). The curves, despite the noticeable growth, did not take the shape characteristics of the stationary bacterial culture. Complexes of cobalt(III) and sensor BsOx are colored compounds, so may negatively affect the measurement result (inner filter effect, deviations from the Beer–Lambert law) [26]. The color of the compound or sample turbidity may affect the size of the phosphorescence signal [3]. Moreover, two ionic compounds through interaction may affect the obtained phosphorescence intensity values influencing the obtained phosphorescence values, as well as binding to cells and sample components. A self-quenching in solution is also possible, which is thought to be partially due to dynamic (collisional) quenching [27]. The sensitivity is limited by optical interferences (light scattering, auto phosphorescence, sample effects, etc.), causing relatively high and variable background signals [28].

#### 2.3.2. Measurements Using Box and BsOx in 5% and 50% Bovine Serum Albumin (BSA)

BSA is a homolog of HSA (human serum albumin), which is an abundant plasma protein that binds a wide range of drugs and metabolites and is involved in the transport of metal ions and metal complexes with drugs through the bloodstream. Binding to these proteins may lead to the loss or enhancement of the biological properties of the original drug, or provide paths for drug transportation. Kudva et al. [29] showed docking results that indicate that both human and bovine serum albumins have only one favorable binding site each for AmB. Due to its low cost and highly structural homology with HSA, this represents the serum albumin that is preferred in laboratory practice for the experimental and theoretical studies of protein-drug interactions [30].

To evaluate the influence of albumin on the activity of the studied compounds, tests were carried out in the medium RPMI 1640 with the addition of bovine serum albumin (BSA) in the amount of 5 and 50% for these same strains of *Candida albicans*. The investigation of the binding interaction between complexes and serum albumin is the first step towards clarifying the detailed understanding of the pharmacology of drugs [31].

We have shown in our studies that the presence of BSA contributed to a significant reduction in the activity of Co(III) complexes (Table 2).

In 5% BSA, the MIC values of compound (**1**) increased eight-fold compared to MIC without BSA for both strains of *Candida albicans*, and in the case of compound (**2**) the MIC values reached a value greater than 1000 µg/mL. In samples containing 50% BSA, the MIC values for compound (**1**) increased 16-fold compared to the samples without BSA, and for compound (**2**), it exceeded the concentration of 1000 µg/mL for both *Candida albicans* strains. However, the MIC_s_ of AmB in the presence of 5% and 50% bovine serum albumin (BSA) decreased two-fold, from 0.5 to 0.25 µg/mL, for both strains of *Candida albicans*.

The studies with AmB conducted by Zhanel et al. [32], Kudva et al. [29], and Senrviratne et al. [33] showed no increase in MIC value in the presence of human serum (HSA) and bovine serum albumin (BSA) in the case of AmB, which is not consistent with the free-drug hypothesis [34]. According to this hypothesis, the free (unbound) drug rather than the protein-bound drug is assumed to be active, since with a drug avidly bound to serum, protein activity declines in the presence of plasma. However, Schaffer-Korting et al. [35] observed that amphotericin, miconazole, and griseofulvin activity declined in the presence of albumin. Amphotericin B has a binding region on human and bovine serum albumin; this property enables the conversion of a water-insoluble substance to a soluble and transportable one [36]. Does albumin, as in the case of amphotericin B, have binding sites for this type of compound, or are these interactions non-specific with albumin, leading to a decrease in the activity of Co(III) compounds against the tested *Candida albicans* strains? The answer to these questions will be one of the topics of our next article.

The most important property of albumin is its ability to reversibly bind with various ligands, such as cysteine, glutathione, pyridoxal phosphate, and Schiff base ligands linked to metal ions (Co(II), Cu(II), Mn(II), Ni(II), Hg(II), Zn(II)) [37]. From the work of Pontoriero et al. [38], it is known that the Co(III)-sulfathiazole complex ([Co^III^(stz)_2_OH(H_2_O)_3_]) partially quenched the native fluorescence of bovine serum albumin (BSA), which suggests its specific interaction with the protein. Additionally, Vignesh et al. [39] in their studies showed a strong interaction between double-chain surfactant–cobalt(III) complex and BSA.

In our study, using phosphorescence optical respirometry, we showed that the MIC value of AmB in the presence of 5% bovine serum albumin (BSA) did not change in probe with Box against the reference strain of *Candida albicans* (0.5 µg/mL), whereas in samples containing the clinical strain, the MIC value dropped to 0.25 µg/mL (Figure 4a,b). In the case of samples where the BsOx was used, a four-fold increase in the activity of AmB (MIC values decreased from 0.5 to 0.125 µg/mL for both strains of *C. albicans*) was observed (Figure 4a,b). The increase in the activity of AmB may result from (a) the synergistic interaction of the sensor with AmB, (b) AmB and BSA binding and easier *Candida albicans* cell penetration, or (c) the phenomenon of an increase in activity as a result of excitation with light of a specific wavelength (photodynamic therapy).

The antifungal activity of Co(III) complexes with diamine chelate ligands in the presence of 5 and 50% BSA decreased significantly for both sensors. The only exception was the samples in the presence of 5% BSA where Box was used as the sensor, in which the MIC values were unchanged for compound (**1**) and were 62.5 µg/mL for both *C. albicans* strains. For the same sensor, the MIC_S_ of compound (**2**) were 250 µg/mL against both tested strains of yeasts, thus the value increased four-fold (from 62.5 to 250 µg/mL). The MIC_S_ of compounds (**1**) and (**2**) were 250 µg/mL in the case of the reference strain of *C. albicans*, and 500 µg/mL for compound (**2**) against the clinical strain of *C. albicans*, for samples containing BsOx. However, in samples containing 50% BSA, the MIC_S_ of compounds (**1**) and (**2**) had the same value using the Box sensor in both strains of *Candida* spp., which were 250 and >1000 µg/mL, respectively. MIC values in the presence of a soluble BsOx sensor for both strains of *Candida albicans* increased four-fold, from 62.5 ug/mL to 250 ug/mL (Figure 5a,b). When analyzing the activity of AmB in the presence of 50% BSA and using the BsOx sensor, a drastic decrease in its activity against *C. albicans* ATCC 10231 (MIC > 4 ug/mL) and a two-fold decrease in activity against *C. albicans* 12823 was observed. However, it needs to be emphasized that the curve shapes describing the growth of fungi in the conditions of 50% BSA are very distorted and differ drastically from the typical sigmoidal shape of the curves (Figure 5a,b). Various protein fractions in BSA can affect the binding of the drug and thus its antifungal activity. We hypothesize that BsOx largely interfered with BSA, reducing its phosphorescence intensity signals and making the calculation of the MIC values inaccurate. It is also possible that compounds can effectively block the penetration of excitation causing a strong ‘inner filter effect’ on probe excitation and light into the sample. Additional optical effects come into play, namely prominent light scattering. These spectral effects can strongly influence the intensity signals of the sensor [26]. Moreover, BSA tends to act as a scavenger of the reactive species when it is present in the reaction medium [27].

The sigmoidal shape of the curves obtained during respirometric analyzes reflects the process of deoxidation of the sample, which was determined by the initial number of microorganisms and their proliferation rate. In some cases, complete deoxygenation of the sample was not observed, indicating that the yeast was dying or respiration was switching to anaerobic metabolism when the oxygen concentration became too low. When cells are exposed to the toxic action of chemical compounds, their viability decreases, which may affect the magnitude of the phosphorescence signal. The shift in time of the logarithmic growth phase (intense phosphorescence increase) and the different shapes of the curves indicates a slowdown in the growth rate of yeast as a result of the toxic action of chemical compounds. Optical respirometry based on phosphorescence, therefore, allows the monitoring of microorganisms that are still alive (breathing), but not growing, which is impossible to observe in other methods of analyzing cell viability. The results of analyzes of the antimicrobial activity of chemical compounds present in the paper seem to confirm the above observations. It is worth mentioning that the measurements, even in the case of slow-growing strains with low initial culture density, can be measured in less than 24 h.

## 3. Materials and Methods

### 3.1. Strains, Media, and Growth Conditions

The reference and clinical strains of *Candida albicans* ATCC 10231 and *Candida albicans* 12823 were derived from the collection of the Department of Pharmaceutical Technology and Biochemistry, Faculty of Chemistry, Gdansk University of Technology. Strains were stored as glycerol stock at −70 °C. For research purposes, cultures of *Candida* spp. were conducted at 28 °C in RPMI 1640 broth (Sigma-Aldrich, Poznań, Poland).

### 3.2. Chemicals, Methodology, and Apparatus

Amphotericin B and the remaining precursors were purchased as commercially available chemicals from Sigma-Aldrich (Merck). These compounds were used for the synthesis of ruthenium oxygen biosensors (BsOx) as well as (Box) without further purification or drying. The *tris*-(4,7-diphenyl-1,10-phenanthroline)ruthenium(II) chloride pentahydrate [Ru(DPP)_3_]Cl_2_ x 5 H_2_O adsorbtion on the DavisilTM silica gel, its embedding in the silicone rubber Lactite NuvaSil^®^ 5091, and its coating on the bottom of 96-well plates (Box), was performed according to the procedure described elsewhere (Hałasa et al., [12]). [Ru(DPP)_3_]Cl_2_ = Box immobilized on silica gel is insoluble in water in this form, contrary to the ruthenium complex [Ru[DPP(SO_3_Na)_2_]_3_]Cl_2_ = BsOx. The percentage compositions of elements of the coordination compounds synthesized in solid forms were determined using an element analyzer Carlo Erba EA CHNS. The infrared spectra of all compounds studied were recorded in the wavenumber range 4000–400 cm^−1^ on a Spectrum Two IR instrument (Perkin Elmer) using the ATR technique. NMR spectra of coordination compounds dissolved in D_2_O were registered with a Brücker AVANCE III 700 MHz spectrometer. UV-Vis absorption spectra of the BsOx and Co(III) complexes’ aqueous solutions were recorded on an Evolution 300 spectrophotometer (ThermoScientific) with a data interval of 1 nm, a slit width of 1.0 nm, and a scan rate of 240 nm/min. Determination of the purity of the ruthenium(II) complex oxygen sensor was performed using analytical UHPLC on a Thermo Scientific Vanquish UV-Vis DAD detector and an Accucore C18 column (2.6 μM, 150 mm × 3.0 mm) at 25 °C and a flow rate of 0.2 mL/min, with detection at 438 nm. Mobile phases were mobile phase A, HPLC grade acetonitrile; and mobile phase B, 1% formic acid in dH_2_O. The following mobile phase gradient was used: 5% A (containing 95% B) from 0 to 3 min; 5–95% A (95–5% B) from 3 to 13 min; 95% A (5% B) from 13 to 18 min; 95–5% A (5–95% B) from 18 to 19 min; and 5% A (95% B) from 19 to 24 min. High-resolution electron-spray ionization mass spectrum (HRMS-ESI) of the BsOx aqueous sample was recorded on an Agilent Technologies 6210-1210 TOF-LC-ESI-MS instrument operating in the positive and negative ion modes (two times). Additionally, the MALDI-TOF mass spectra of the Ru(II) and Co(III) complexes studied were registered on an autoflex TOF/TOF maX instrument (Brücker Daltonics) with the DHB matrix. The thermogravimetric analysis was performed using a thermal equalizer TGA 8000 Perkin Elmer coupled with a Spectrum Three instrument (Perkin Elmer). The analyzer was equipped with a programmable temperature controller, which automatically maintains a constant temperature during thermal events. The TG weight-loss measurements were performed in the 28−1000 °C temperature range at a heating rate of 10 °C/min in an alumina crucible (for sample weights 4.574 and 1.607 mg, respectively). All experiments were carried out in an argon atmosphere and verified at least twice. The BsOx cyclic voltammogram (CV) was obtained using an Autolab PGSTAT204 potentiostat/galvanostat (Metrohm Autolab B.V., Utrecht, The Netherlands), controlled by Nova software at room temperature (25 °C). In the three-electrode system, a glassy carbon electrode (GCE; 2 mm) served as the working (polarizable) electrode, an Ag/AgCl electrode (SCE) as the reference electrode (non-polarizable), and a platinum wire (Pt) as the counter electrode. The custom-made GCE (mineral) was cleaned before analysis with 0.2 µm Al_2_O_3_. Non-aqueous solutions of {Ru[DPP(SO_3_Na)_2_]_3_}Cl_2_ with a volume of 5 mL had a concentration of 10^−3^ M (DMSO), and the standard electrolyte used for the non-aqueous sample tetrabutylammonium perchlorate (TBAP, 0.1M) was added. The BsOx-DMSO solution was deoxygenated (2 min) with pure argon before the electrochemical analysis. The conditions preventing the formation of the “competing peaks” related to the oxygen presence inside the sample were preserved [40] and the CV investigation was carried out at a scan rate of 100 mV/s.

### 3.3. The Synthesis of the Tris-[(4,7-Diphenyl-1,10-Phenanthrolinedisulphonic Acid Disodium) Ruthenium(II)] Chloride Hydrate (BsOx)

The synthetic pathway of oxygen biosensors such as the ruthenium(II) with a DPP(SO_3_Na)_2_ complex was previously reported in the literature [22,41]. The solid precursors of RuCl_3_ trihydrate (16 mg, 0.0623 mmole) and solid 4,7-diphenyl-1,10-phenanthroline disulfonic acid disodium salt trihydrate (130 mg, 0.220 mmol) were dissolved in 40 mL of distilled water, and the mixture was refluxed with stirring for two days (the stoichiometry of Ru(II) ion: ligand was 1: 3.5 eq.). Then, the mixture was cooled to room temperature and filtrated by using the 0.22 µm PTFE syringe filter and the rotary evaporation yielded the solid crude product. The sample obtained was dissolved again in the minimum volume of the distilled water and passed down with a Sephadex LH-20 (33 × 2 cm) column (see Appendix A). The distilled water was used as an eluent to separate the complex isomers. The second red fraction, obtained after the first green fraction was discarded, turns out to be the expected main isomer, called the oxygen biosensor BsOx (Appendix A). The expected red isomer was dried for the next 6 h in a vacuum at 30 °C to yield 55 mg (41%) of a deep red, amorphous substance. Note that the product analyzed showed its hydration state due to the use of other precursors than those reported in [21,42]. The elemental analysis of solid BsOx allows us to determine the formula RuC_72_H_86_N_6_Na_6_S_6_O_40_Cl_2_: found (calc): %N: 3.80 (3.85); %C: 39.18 (39.67); %H: 3.34 (3.94); %S: 8.70 (8.81). Note that the standard solid drying procedure, routinely used before this type of analysis, was omitted consciously in the case of the BsOx sample provided. BsOx ATR signals [cm^−1^] obtained proved the hydrated state of BsOx solid form synthesized: 3410.04 ν(OH); 3110.02 ν(CH); 1614.84 ν(C=C); 1590.97 ν(C=N); 1529.97 ν(C=C); 1466.75 δ(CH); 1411.96 ν(-SO_2_-)_as._; 1122.08 δ(OH); 1095.44 and 1029.62 (CH)_aroma_; and 406.80 ν(RuN) (Appendix A). The BsOx ^1^H NMR spectrum and chemical shifts were (500 Hz; D_2_O); δ [ppm]: 9.08 (d, 2H) J = 5.0 Hz; 8.23 (d, 2H) J = 5.48; 8.07 (s, 2H); 7.95–7.85 (m, 4H); and 7.69–7.64 (m, 4H) (Appendix A). LCMS chromatograms as a result of time-period deconvolution (Appendix A) together with the proper MALDI spectrum of BsOx (Appendix A) are introduced in the Appendix A. The molecular weight (M.W.) of BsOx was confirmed based on LCMS and MALDI results. The predicted M.W. of {Ru[DPP(SO_3_Na)_2_]_3_}Cl_2_ is 1781.5 g/mol and the [M + Na^+^ + H^+^ ] = 1805.5 (*m*/*z*, MS positive mode) and [M − Na^+^]^−^ = 1758.5 (*m*/*z*, MS negative mode) were found in LCMS. All experimental data obtained are in good agreement with the data reported previously [21,22,41].

### 3.4. The Synthesis of the Tris-[(4,7-Diphenyl-1,10-Phenanthroline) Ruthenium(II)] Chloride Pentahydrate (Box)

The synthetic pathway of the oxygen biosensor—the ruthenium(II) with DPP complex—was previously reported in the literature [12]. The ruthenium(II) phenanthroline complex was synthesized according to a modified method [42]. Ruthenium(III) chloride (0.5 mmol) was dissolved in water (0.25 mL) and ethylene glycol (3 mL) and then 4,7-diphenyl-1,10-phenanthroline (1.5 mmol) was added. The mixture was refluxed for 5 min under microwave irradiation and cooled down before the addition of brine and chloroform. The organic layer was concentrated under reduced pressure. The residue was recrystallized from ethanol-water (2:1) to yield [Ru(DPP)_3_]Cl_2_ (475 mg, 75%) as an orange powder.

Box ATR signals [cm^−1^] obtained proved the hydrated state of the Box solid form synthesized: 3376.18 ν(OH); 3056.93 ν(CH); 1621.34 ν(C=C); 1595.00 ν(C=N); 1558.44 ν(C=C); 1492.00–1444.03 δ(CH); 1176.13 δ(OH); 1091.75 and 1020.14 (CH)_aroma_; and 462.97 ν(RuN) (Appendix A). The Box compound presents MLCT absorption bands in the visible region at about 440 and 465 nm (Appendix A) and also the π → π^*^ transitions centered on the ligand in the UV range. The absorption band at 278 nm is associated with the presence of the 4,7-diphenyl-1,10-phenanthroline ligand. The molecular weight (M.W.) of Box was confirmed based on MALDI results; *m*/*z* signals found (calc.): [M + H] 1170.38 (1170.24); [M-2Cl] 1098.37 (1098.24); the charges of deionized forms of Box were omitted (Appendix A). All experimental data obtained are in good agreement with the data reported previously [12].

### 3.5. The Synthesis of Compounds *(**1**)* and *(**2**)*

The synthesis of [CoCl_2_(dap)_2_]Cl (**1**) and [CoCl_2_(en)_2_]Cl (**2**) were performed according to the procedure modified by Chylewska et al. [43,44] which was preliminarily reported by Bailar et al. [45]. A volume of 4 mL of 99% solution of ethylenediamine (en) was added, with stirring and heating, to a solution of 12 g of cobalt chloride hexahydrate in 30 mL of water in a 100 mL bottle. In the next step, 4 mL of 30% H_2_O_2_ was added dropwise to the mixture, and a vigorous stream of air was drawn for 30 min. After this time, 20 mL of 36% HCl solution was added and the probe was heated for the next 2 h by using a water bath to obtain the final reduced volume of 30 mL. The resulting system was cooled down and placed in the refrigerator. The next day, the green crystals of [CoCl_2_(en)_2_]Cl (**2**) were filtered off and washed with ice-cold water to give the product expected. The same procedure was incorporated in the case of [CoCl_2_(dap)_2_]Cl (**1**) compound synthesis with the ligand exchange; the ethylenediamine was replaced by 5 mL, 99% 1,3-diamine propane solution (dap). The additional crystallization process of the powder form of (1) from absolute methanol was conducted. The structural characterizations of both Co(III) complexes were made afresh due to the resynthesis performed. Compound (**1**) structure confirmation: ATR signals of crystals (1) [cm^−1^]: 3180 ν(NH_2_); 2978 (CH_alif._); 1632 δ(NH_2_) and 1553 ρ(C-N); 1297 (C-H _alif._) and 1041 ν(C-C); 665-500 cm^−1^ (*trans- form*), 463 and 410 ν(CoN)—Appendix A. The MALDI-TOF data obtained for (1) (*m*/*z* signals) [M] found (calc.) [M + 6H + 2Na] 366.037 (366.54); [M + 3H] 316.354 (316.54); [M-4H-HCl] 273.049 (273.54); [CoH_4_L] 136.997 (137.54); [LH] 75.106 (75.00); the charges of deionized forms of (1) were omitted, see Appendix A. The UV-Vis spectrum of (1)–maxima positions and description [nm]: 304; 468 and 511 (d-d MLCT)—Appendix A. Elemental analysis results obtained for [CoCl_2_(dap)_2_]Cl [6HCl] powder [found (calc.)]: %C 13.63 (13.52); %H 4.617 (4.88); %N 10.80 (10.51). The second elemental analysis results obtained for [CoCl_2_(dap)_2_]Cl [1.5HCl] crystals [found (calc.)]: %C 19.48 (19.55); %H 5.62 (5.84); %N 15.08 (15.20). The COSY, ^1^H, ^13^C, and HSQC NMR spectra obtained for compound (1) were included in the Appendix A together with their elaboration (Appendix A). Compound (**2**) structure confirmation: ATR signals of crystals (2) [cm^−1^]: 3219 and 3125 ν(NH_2_); 2909, 2800 and 2046 ν(CH_alif._); 1573 δ(NH_2_) and 1499 ρ(CN); 1119 ν(CH _alif._) and 1051 ν(C-C); 813 (CH_2_ rocking); 630-500 cm^−1^ (*trans- form*), 464 and 419 ν(CoN)—Appendix A. The MALDI-TOF data obtained for (2) (*m*/*z* signals) [M] found (calc.) [M + H] 286.035 (286.48); [M-H-H_2_L] 223.16 (223.48); [MH-H_3_L] 223.163 (223.04); the charges of deionized forms of (2) were omitted, see Appendix A; UV-Vis spectrum of (2)—maxima positions and description [nm]: 404; 458 and 620 (d-d MLCT)—Appendix A. Elemental analysis results obtained for [CoCl_2_(en)_2_]Cl [found (calc.)] %C 17.17 (16.83); %H 6.237 (5.65); %N 19.98 (19.63). The COSY, ^1^H, ^13^C, and HSQC NMR spectra obtained for compound (**2**) were included in the Appendix A together with their elaboration (Appendix A).

### 3.6. Study of the Effect of Co(III) Coordination Compounds and Amphotericin B on Yeast by Phosphorescence Optical Respirometry

#### 3.6.1. Measurements Using Box and BsOx in RPMI Medium

The studies with Box were performed according to the procedure described in the paper of Hałasa et al. [12], whereas the tests with BsOx were carried out as follows. Eighteen-hour-old yeast cultures were diluted 10-fold with RPMI broth and incubated for 2–3 h with shaking. Then, the samples were prepared in 96-well black flat microtiter plates. The wells were filled with 150 µL of the suspension of *C. albicans* strains in growth medium (RPMI 1640) (0.5 McFarland and 0.5 McFarland diluted 10×, 100×, 1000×, 10,000×, and 1,000,000×) and 10 µL of the oxygen sensor (in the range of 7.8–250 µg/mL). The control samples contained growth medium, growth medium with BsOx, and growth medium with yeast suspension and sensor. On the surface of the tested samples, 100 μL of mineral oil was applied and the microtiter plates were measured at 28 °C with shaking. All measurements were performed in the microplate reader Infinite M200 Pro (Tecan, Switzerland). The band-pass filters used for phosphorescence measurements were 480 nm for excitation and 610 nm for emission. The phosphorescence readings were normalized by division of the results obtained every 10 min of the experiment by the results for the control, with no bacterial cells added as the phosphorescence readouts were in arbitrary units. Phosphorescence was measured from the top of the microtiter plate.

#### 3.6.2. Measurements Using Box and BsOx in 5% and 50% Bovine Serum Albumin (BSA)

In studies with Box, 80 μL of the proper solution was introduced to the wells of the microtitre plate coated with the oxygen biosensor: Co(III) complexes with diamine chelate ligands (500–15.6 μg/mL) and AmB (4–0.06 μg/mL). The microbial suspension (80 μL) in the amount of 10^5^ CFU/mL was added to each well. On the surface of the test samples, 100 μL of liquid paraffin was applied and the plate was placed in a plate reader. The concentration range of compounds used in the experiment was determined from previously performed tests using the serial dilution method. All experiments were performed in triplicate.

The samples for studies with BsOx were prepared in 96-well black flat microtiter plates. The wells were filled with 70 µL of the suspension of the *C. albicans* strains in growth medium (RPMI 1640, about 10^5^ CFU/mL), 10 µL of the oxygen sensor (62.5 µg/mL), and 80 µL of the tested compounds. The control samples contained growth medium, growth medium with BsOx, and growth medium with a yeast suspension + sensor. On the surface of the tested samples, 100 µL of mineral oil was applied and the microtiter plates were measured at 28 °C with shaking. All measurements were performed in the microplate reader Infinite M200 Pro (Tecan, Switzerland). The band-pass filters used for phosphorescence measurements were 480 nm for excitation and 610 nm for emission. The phosphorescence readings were normalized by division of the results obtained every 10 min of the experiment from the results for the control with no bacterial cells added, as phosphorescence readouts were in arbitrary units. Phosphorescence was measured from the top of the microtiter plate.

### 3.7. Determination of the Minimum Inhibitory Concentration (MIC) in RPMI Medium and Bovine Serum Albumin

#### 3.7.1. Determination of the Minimum Inhibitory Concentration in the RPMI Medium

The MIC of the tested compounds was determined by the microbroth dilution method using 96-well plates according to the Clinical and Laboratory Standards Institute [46], against the selected reference and clinical strains of yeast of the genus *Candida*. Dry test samples of amphotericin B were dissolved in DMSO (dimethyl sulfoxide) to give a stock concentration of 6400 µg/mL. Crystals of Co(III) compounds were dissolved in sterile water to give a stock concentration of 32,000 µg/mL. The culture of yeasts was prepared by transferring cells from the stock cultures to tubes with RPMI 1640 broth incubated with agitation for 24 h at 28 °C. The cultures were diluted with the same broth to achieve an optical density corresponding to 10^3^ colony-forming units per mL (CFU/mL). After filling each well with 100 µL of broth water, solutions of tested coordination compounds were added to the first well of each microtiter line. Dilution in geometric progression was performed by transferring the dilution (100 µL) from the first to the twelfth well. An aliquot in the volume of 100 µL was discarded from the twelfth well. In the case of DMSO solutions, 96 µL of RPMI 1640 broth and 4 µL of the appropriate concentration of antifungal drug were added to each well. The fungal suspension (100 µL) at 10^3^ CFU/mL was added to each well. The final concentration of the tested compounds ranged from 128 µg/mL to 0.0625 µg/ mL (DMSO solutions) and from 8000 µg/mL to 3.9 µg/mL (water solutions). Plates were incubated at 28 °C for 24 h and then fungal growth was assessed. The MIC was taken as the lowest sample concentration that prevented visible growth [12].

#### 3.7.2. Determination of the Minimum Inhibitory Concentration in Bovine Serum Albumin

The tests were carried out according to the procedure described above, except that 5 and 50% bovine serum albumin was added to the RPMI medium.

### 3.8. Statistical Analysis

All experiments were carried out in triplicates, in three independent experimental sets.

The means ± SD were used in the data and graphics’ statistical analysis.

## 4. Conclusions

In this study, Ru(II) based phosphorescent oxygen sensors in the form of ruthenium-*tris*-(4,7-diphenyl-1,10-phenanthroline) dichloride (Ru(DPP)_3_Cl_2_ (Box) adsorbed on the Davisil^TM^ silica gel and then embedded in the silicone rubber Lactite NuvaSil^®^ 5091 and the coating on the bottom of 96-well plates), and the water-soluble *tris*-[(4,7-diphenyl-1,10-phenanthroline disulphonic acid disodium)ruthenium(II)] chloride hydrate {Ru[DPP(SO_3_Na)_2_]_3_}Cl_2_, BsOx) were used. BsOx was synthesized and subsequently analyzed by RP-UHPLC, LCMS, MALDI, elemental analysis, ATR, UV-Vis, ^1^H NMR, and TG/IR techniques. The results obtained in this study indicate that both Ru(II)-based oxygen sensors can be successfully used to test the antifungal activity of various chemical compounds and determine the metabolic state of organisms under various conditions. Furthermore, when comparing both sensors, better and more unambiguous results can be obtained with a sensor using silica-adsorbed Ru(II), embedded in silicone, and immobilized in the wells of a 96-well plate. Embedding it in polymer and immobilization in the wells provide possibilities for the exclusion or minimalization of toxicity and invasive perturbation of biological systems. For the first time, a soluble Ru-based sensor {Ru[DPP(SO_3_Na)_2_]_3_}Cl_2_), herein called BsOx, was used to test the antifungal activity of chemical compounds against *Candida albicans* strains using the phosphorescence optical respirometry method. Moreover, this is the first time that the two types of biosensors were used to observe the effects directly related to Co(III) diamine complexes studied. It was also shown for the first time in this work that BsOx enhanced the activity of the antifungal drug, in this case, amphotericin B, causing its activity against *Candida albicans* to increase significantly, which may indicate a synergistic effect of the compounds mentioned. Simultaneously, the same sensor caused a decrease in the activity of the Co(III) complexes, suggesting an antagonistic effect. These results, in relation to BsOx, will be our next research goal. It should be noted that there are no literature data to evaluate the synergism of the tested chemical compounds with phosphorescence optical respirometry. We hypothesize that through demonstrating either the synergistic effect of the soluble Ru(II)-based sensor-AmB) or the antagonistic effect (Co(III) complexes-soluble Ru(II)-based sensor), it is possible to determine the probable mechanism of action of the compounds. In the case of the Ru(II) complex-AmB mixture, AmB is known to act on a cell membrane component (ergosterol), suggesting that the Ru complex may strengthen the effect of the cell membrane components to increase membrane permeability. Similar to the Co(III)-Ru(II) mixture, a decrease in the MIC value for Co(III) compounds in the presence of the sensor may indicate that both compounds may compete with each other for the binding site. Our future goal is to conduct such studies for a broader spectrum of *Candida* species and antifungals.

## Figures and Tables

**Figure 1 ijms-24-08744-f001:**
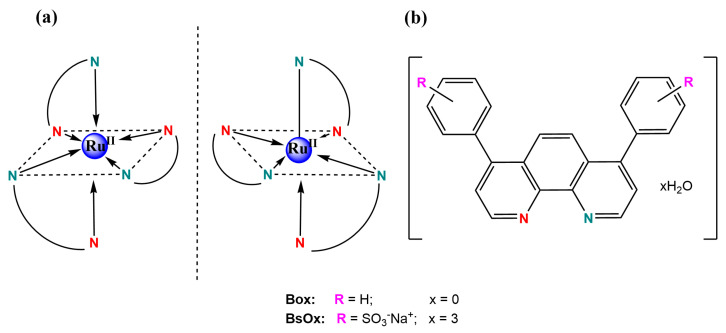
(**a**) Proposed structures of both oxygen biosensor complexes with ruthenium(II); (**b**) both ligands structures with both N, N’-donor atoms marked by different colors.

**Figure 2 ijms-24-08744-f002:**
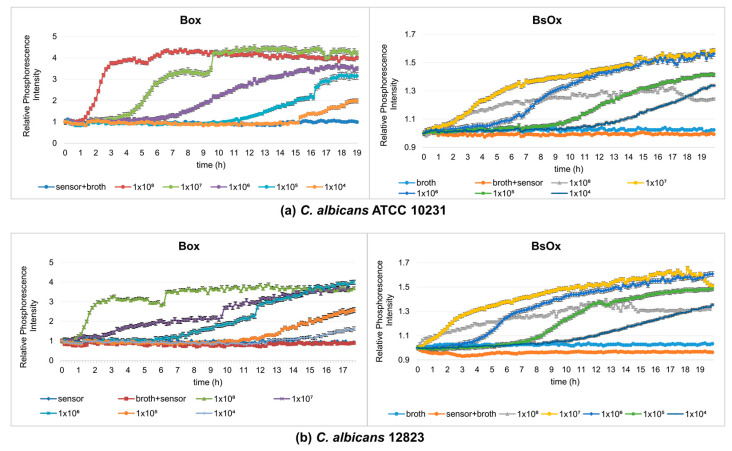
Profiles of relative phosphorescence intensities for different densities of yeast cultures of *C. albicans* ATCC 10231 and *C. albicans* 12823 against time. Measurements using a sensor: (**a**) [Ru(DPP)_3_]Cl_2_ = Box, (**b**) [Ru[DPP(SO_3_Na)_2_]_3_]Cl_2_ = BsOx; concentrations of both sensors used in studies were 62.5 µg/mL.

**Figure 3 ijms-24-08744-f003:**
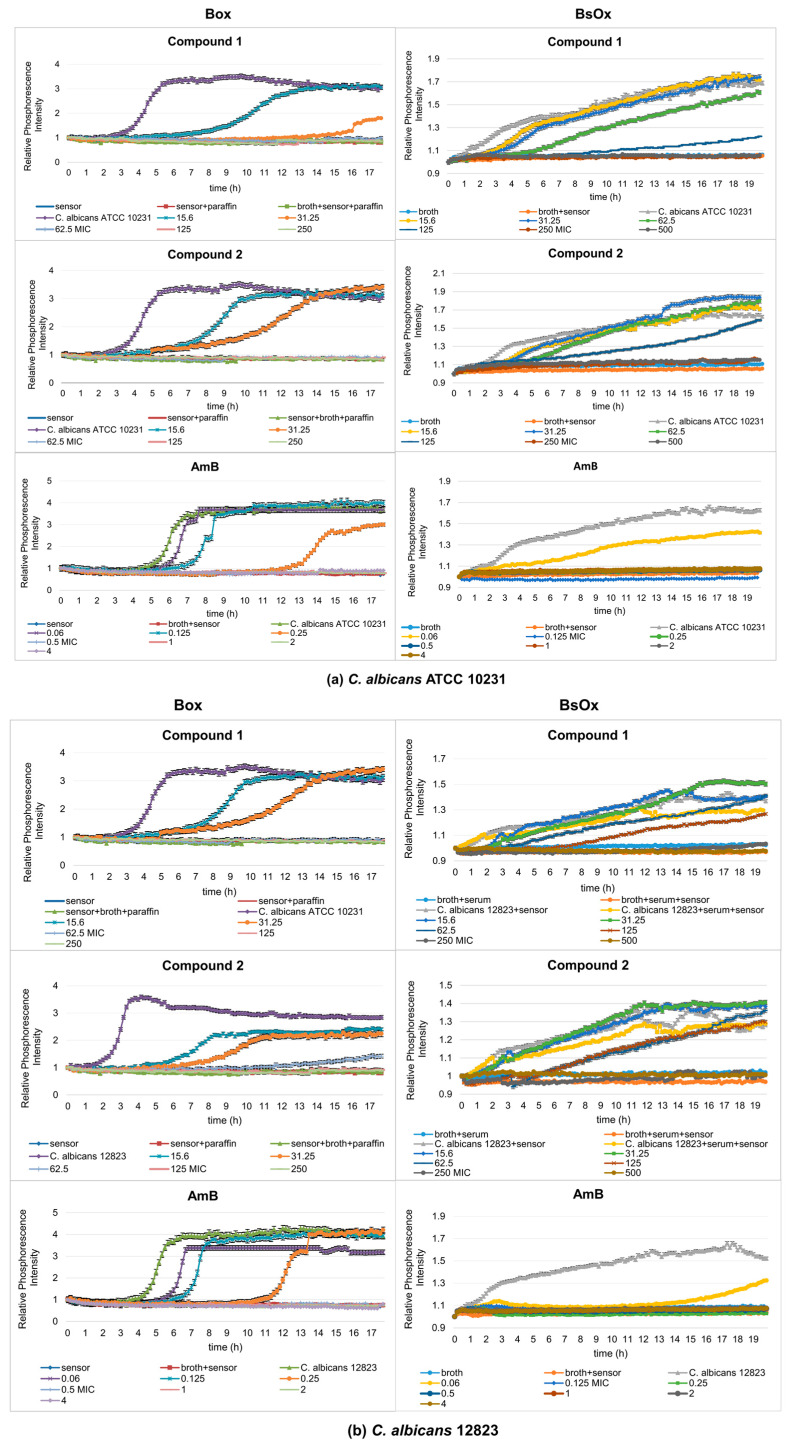
Profiles of relative phosphorescence intensities of (**a**) *C. albicans* ATCC 10231 and (**b**) *C. albicans* 12823 strains against time for different concentrations of compounds (**1**), (**2**)**,** and AmB with Box and BsOx (62.5 µg/mL) sensors. In the legend, all concentrations are expressed in µg/mL.

**Figure 4 ijms-24-08744-f004:**
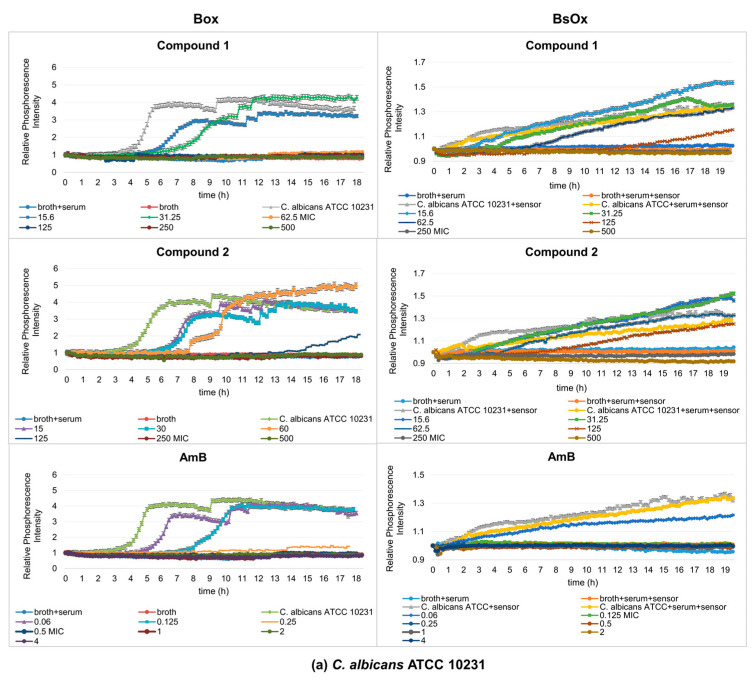
Profiles of relative phosphorescence intensities of the (**a**) *C. albicans* ATCC 10231 and (**b**) *C. albicans* 12823 strains against time for different concentrations of compounds (**1**)**,** (**2**)**,** and AmB with Box and BsOx (62.5 µg/mL) sensors in an environment of 5% BSA. In the legend, the sample concentration is expressed in µg/mL.

**Figure 5 ijms-24-08744-f005:**
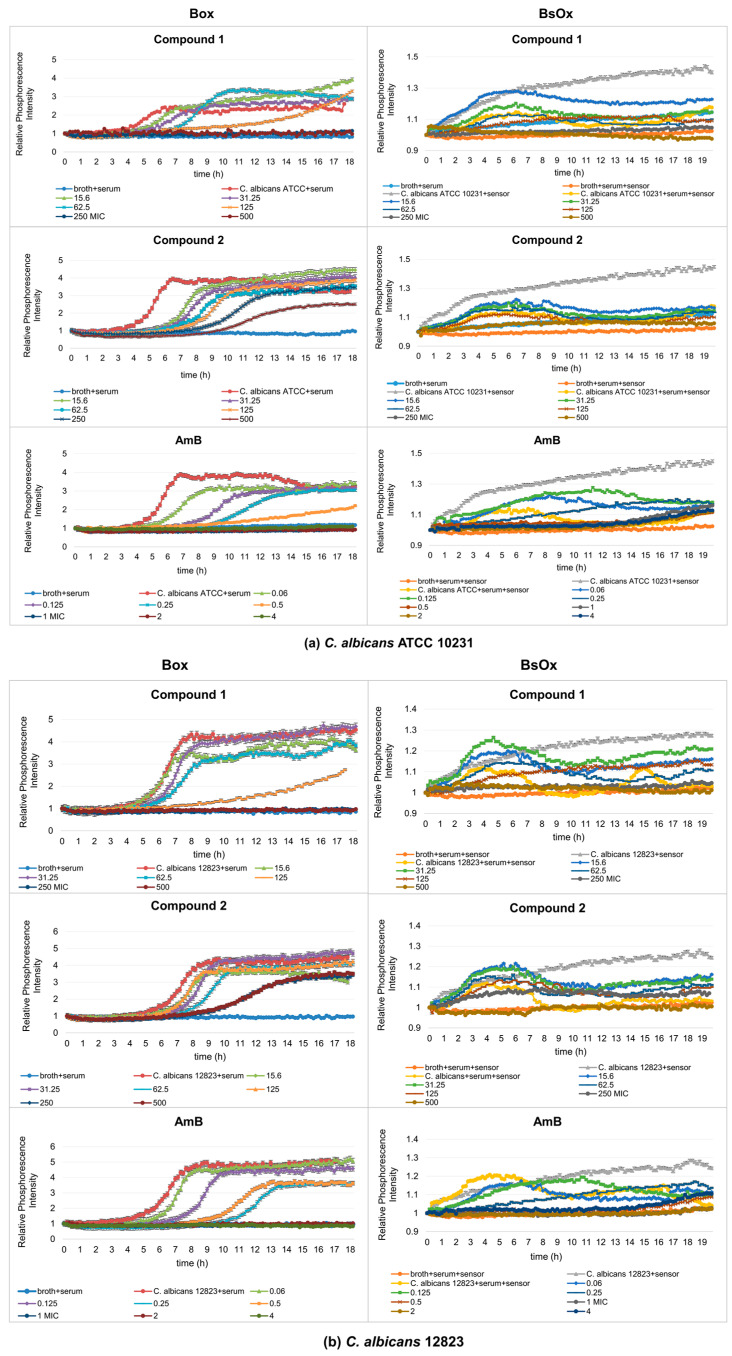
Profiles of relative phosphorescence intensities of (**a**) *C. albicans* ATCC 10231 and (**b**) *C. albicans* 12823 strains against time for different concentrations of compounds (**1**), (**2**)**,** and AmB with Box and BsOx (62.5 µg/mL) sensors, in an environment of 50% BSA. In the legend, the sample concentration is expressed in µg/mL.

**Table 1 ijms-24-08744-t001:** MIC values in μg/mL of [CoCl_2_(dap)_2_]Cl (**1**), [CoCl_2_(en)_2_]Cl (**2**), and AmB against *Candida* spp. strains (serial dilution method).

Strains	MIC of (1)	MIC of (2)	MIC of AmB
*C. albicans* ATCC 10231	62.5 ± 4.93	62.5 ± 4.01	0.5 ± 0.06
*C. albicans* 12823	62.5 ± 4.01	62.5 ± 4.92	0.5 ± 0.06

**Table 2 ijms-24-08744-t002:** MIC values in μg/mL of [CoCl_2_(dap)_2_]Cl (**1**), [CoCl_2_(en)_2_]Cl (**2**), and AmB in 5% and 50% BSA against *Candida* spp. strains (serial dilution method).

Strains	MIC of (1)		MIC of (2)		MIC of AmB	
	5% BSA	50% BSA	5% BSA	50% BSA	5% BSA	50% BSA
*C. albicans* ATCC 10231	500 ± 112	1000 ± 224	>1000	>1000	0.25 ± 0.06	0.25 ± 0.07
*C. albicans* 12823	500 ± 137	1000 ± 274	>1000	>1000	0.25 ± 0.07	0.25 ± 0.07

## Data Availability

The data presented in this study are available on request from the corresponding authors.

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
