# Peer review of "Ru(II) Oxygen Sensors for Co(III) Complexes and Amphotericin B Antifungal Activity Detection by Phosphorescence Optical Respirometry"

_ijms, 2023, doi:10.3390/ijms24108744_

Round 1

Reviewer 1 Report

Dear Author,

you can find attached my list of major revision. I hope that you can explain clearly the novelty and the importance of the research in this manuscript. You can find also some problems related to the style. In particular the list spectroscopical data (frequency peaks) must be reported in the experimental section, not in the Results and discussion.

Reviewer 2 Report

Dear Authors,

You present here the study of Ru(II)-based phosphorescent sensors for the investigation of the activity of Co(III) complexes with diamine chelate ligands and amphotericine B on Candida albicans.

 The first observation that I have is that you need to change the title of the manuscript, because the present one is not very clear and not very well formulated.

Secondly, I suggest an extensive check of English grammar and language. There are many mistakes that need correction, in order that your paper is well written and easy to read and understand. Some examples: 

- add "an' in line 14 of Abstract, in order to be "is an important element"

- rephrase lines 15 and 16

- in line 24 is written "the studies were performed". Which studies? it is not clear.

- correct "yeats" in line 55 into "yeasts"

- add "that" in line 77 : "We showed that these assays..."

- add "the" in line 93: "with the one obtained..."

....

I suggest that you keep only the essential images and the others present them in Supplementary files. I consider that there are too many Figures in the manuscript.

I suggest you detail the synthesis of Co(III) complexes.

Also, the Conclusions part has to be detailed.

The references are well chosen and presented.

The quality of English Language is not satisfactory, you have to make some corrections.

Reviewer 3 Report

The article: Ru(II)-based phosphorescent oxygen sensors for Co(III) complexes with diamine chelate ligands and amphotericin B studies against Candida albicans strains was evaluated for publication in Int. J. Mol. Sci. and rejection of the article is suggested. Ru(II)-based phosphorescent oxygen sensors are well known, described in detail, and have been used in the present work in the field of microorganisms. None of this is new, and the language used is very difficult to understand in some parts. Also, the pictures in the article are always the same, showing only response - profiles of relative phosphorescence.

The language used is very difficult to understand in some parts. 

Round 2

Reviewer 1 Report

Dear author,

thanks for your efforts to improve the manuscript. I think that the work is suitable for publication. 

Author Response

Dear Reviewer,

thank you for the recommendation of our manuscript to publication.

Reviewer 2 Report

Dear Authors, 

Thank you for considering my comments and suggestions and for making the changes required.

Still, from my point of view, the title is not correct from the English language point of view, even if you changed it. 

Still, from my point of view, the title is not correct from the English language point of view, even if you changed it. 

Author Response

Dear Reviewer,

we would like to thank you again for reviewing our manuscript, for your time, valuable and useful comments and remarks. According to the Reviewer’s suggestiontitle the title was changed. We hope that the current version will be correct.

Kind regards

Authors

Reviewer 3 Report

The article was significantly improved and could be published, only the corrections to the pictures are still missing.

Author Response

Dear Reviewer,

we would like to thank you again for reviewing our manuscript, for your time, valuable and useful comments and remarks. According to the Reviewer’s suggestion the corrections to the pictures were entered. We hope that the current version will be correct.

Kind regards

Authors